# Exploring the use of a gamified intervention for encouraging physical activity in adolescents: a qualitative longitudinal study in Northern Ireland

Rekesh Corepal,[1] Paul Best,[2] Roisin O'Neill,[1] Mark A Tully,[1] Mark Edwards,[3] Russell Jago,[3] Sarah J Miller,[2] Frank Kee,[1] Ruth F Hunter[1]

[1]UKCRC Centre of Excellence for Public Health/Centre for Public Health, Queen's University Belfast, Belfast, UK
[2]Centre for Evidence and Social Innovation, School of Social Sciences, Education and Social Work, Queen's University Belfast, Belfast, UK
[3]Centre for Exercise Nutrition and Health Sciences, School for Policy Studies, University of Bristol, Bristol, UK

**Correspondence to**
Dr Ruth F Hunter;
ruth.hunter@qub.ac.uk

## ABSTRACT

**Objective** To explore the temporal changes of adolescents' views and experiences of participating in a gamified intervention to encourage physical activity behaviour and associated processes of behaviour change.

**Design** A qualitative longitudinal design was adopted whereby focus groups were conducted with the same participants in each intervention school (n=3) at four time-points (baseline, end of each of two intervention phases and 1-year follow-up). The framework method was used to thematically analyse the data.

**Setting** Secondary schools (n=3), Belfast (Northern Ireland).

**Participants** A subsample (n=19 at four time-points) of individuals aged 12–14 years who participated in the StepSmart Challenge, a gamified intervention involving a pedometer competition and material rewards to encourage physical activity behaviour change.

**Results** Three core themes were identified: (1) competition; (2) incentives and (3) influence of friends. Participants indicated that a pedometer competition may help initiate physical activity but suggested that there were a number of barriers such as participants finding it '*boring*', and feeling as though they had a remote chance of '*winning*'. 'Incentives' were viewed favourably, although there were participants who found not winning a prize '*annoying*'. Friends were a motivator to be more physically active, particularly for girls who felt encouraged to walk more when with a friend.

**Conclusions** The intervention in general and specific gamified elements were generally viewed positively and deemed acceptable. Results suggest that gamification may have an important role to play in encouraging adolescents to engage in physical activity and in creating interventions that are fun and enjoyable. The longitudinal approach added additional depth to the analysis as themes were refined and tested with participants over time. The findings also suggest that gamified Behaviour Change Techniques align well with core concepts of Self-determination Theory and that various game elements may require tailoring for specific populations, for example, different genders.

**Trial registration number** NCT02455986; Pre-results.

### Strengths and limitations of this study

► A major strength of this study was the novel use of a longitudinal design, using the same participants in repeated focus groups. This enabled the investigation of how participants' views, experiences and physical activity behaviour evolved over time.

► In addition, having a number of researchers involved in the data collection and analyses reduces selectivity and researcher bias.

► However, only three focus groups participated in the study, and all were single-sex schools.

## BACKGROUND

Physical activity (PA) levels in children and adolescents around the world are low.[1] As patterns of PA are established during this time and tend to track into adulthood,[2 3] this is a crucial period that can affect lifelong health and habits. To date, PA programmes for this population have shown limited effect,[4 5] stressing the need for innovative approaches to initiate and maintain PA behaviour.[6]

Programmes such as Pokémon GO illustrates the potential of gamified interventions (ie, the application of elements of game playing, such as scoring points, competing against others, to change behaviour) for encouraging PA behaviour[7] and can provide useful insights into how to reach and engage the most inactive in PA behaviour.[8] Elements of gamification are incorporated into many commercial PA promotion apps, such as Pokémon GO, Fitocracy and 'Zombies, Run!', which include the collection of points for undertaking a targeted behaviour, completing challenges or competing against others in virtual games.[9] Interventions that have applied gamification elements suggest it could be possible to make a routine activity such as travelling to school into a game that

promotes active travel modes and that is engaging and fun.[10 11]

Some key gamification strategies, including feedback on players' performance to allow them to set goals and monitor progression, competing with others and use of incentives, are all evidence-based Behaviour Change Techniques (BCTs).[12] Furthermore, research has demonstrated that other core aspects of gamified interventions such as opportunities for socialisation, self-evaluation and rewarding positive behaviour are key to providing an enjoyable experience,[13 14] and enjoyment has been identified as a significant predictor of PA behaviour.[15] However, gamification interventions have rarely been grounded in well-established theoretical frameworks, and we know little about the views and experiences of participants.

The aim of this study was to explore the views and experiences of adolescents who participated in a gamified PA intervention based on Self-determination Theory (SDT), and the temporal changes of these views and experiences over the 1-year study period. Study objectives included:

1. To explore key aspects of a gamified PA intervention over a 1-year period using a qualitative longitudinal research (QLR) method.
2. To discuss key issues relating to the intervention, such as PA opportunities/barriers, the value of competition and types of rewards and so on.
3. To explore the key influences of PA and to determine who benefited from the intervention, how and why it worked for them.
4. To qualitatively chart changes in behaviours, opinions or views as a result of participating in the intervention.

## METHODS
### Context
The StepSmart Challenge was a 24-week primarily school-based intervention utilising team and individual competitions in five schools in Belfast, Northern Ireland. The school recruitment process is detailed elsewhere (Best *et al*, under review). An independent trial statistician randomly allocated the five schools to the intervention (three schools) or control (two schools) group. School characteristics are shown in table 1: two were all-boys schools, two all-girls schools and one was a co-educational

**Table 1** Characteristics of schools included in the StepSmart Challenge feasibility study

| | Intervention or control group | Single sex or co-educational | Free school meal entitlement (%) |
|---|---|---|---|
| School A | Control | All male | 63.7 |
| School B | Control | Co-educational | 7.2 |
| School C | Intervention | All male | 8.0 |
| School D | Intervention | All female | 56.5 |
| School E | Intervention | All female | 54.6 |

school. All intervention schools were single sex (boys (n=1) and girls (n=2)). Students (n=224) from year 9 classes (aged 12–14 years) were invited to participate in the trial. The main results from the feasibility trial are published elsewhere (Best *et al*, under review). Briefly, the results demonstrated that the StepSmart Challenge was acceptable to young people for encouraging PA, and there was a trend in increasing light-intensity PA and improving mental well-being.

### The StepSmart Challenge
The StepSmart Challenge was a gamified intervention designed according to SDT,[16] using distinct intervention phases aiming to move participants along the motivation continuum from extrinsic motivation towards intrinsic motivation and encouraging PA behaviour change. The theory is grounded in three psychological needs: autonomy, competence and relatedness.[16] Those intrinsically motivated engage in PA for the enjoyment and satisfaction it provides.[17] This form of motivation is associated with improved quality of life, increased PA behaviour[18] and long-term behaviour change.[19] Self-motivation is undermined when individuals feel less control over the activity, and the environment, and if they do not feel a sense of connectedness or belonging to others engaging in the same activity.[20]

Table 2 details the various intervention components and links to BCTs. The intervention consisted of two phases. Phase One involved a multilevel (competition at the school, team and individual levels) pedometer competition lasting for 8weeks. Team selection was determined by the research team and took account of current PA levels and friendship networks measured at baseline; this was to ensure a mixed ability team (4–5 participants per team) with at least one friend in the team. The team competition entailed social incentives such as publication of the results on the website, a trophy awarded to the leading team (ie, the team with the highest number of total steps) in each school at competition end and a prize to the winning school (£1000). During the individual competition in Phase One, material incentives (approximate value of £10; see table 2 for details) were awarded weekly in each school to two participants (participant who accumulated the most steps that week ('Walker of the Week'), and the participant who increased his/her step count the most from the previous week ('Most Improved')). Phase Two (14 weeks) focused on an individual level competition, in which the three participants that had accumulated the most steps in each school during this phase were awarded a 'goody bag' (approximate value of £30 for each participant and consisted of an assortment of those used in Phase One).

### Qualitative longitudinal research
To elicit the temporal views and experiences of participants in the StepSmart Challenge, a QLR design was used involving repeated semi-structured focus groups with the same participants over four data collection periods (recurrent cross-sectional approach). This enabled qualitative charting of the perceived behaviour change as well as changes in attitudes and opinions over a 1-year period.

**Table 2** Intervention components and behaviour change techniques (BCTs)

| Component | Activity/task | BCT (Michie et al, 2013) |
|---|---|---|
| Competition | Competition was designed to take place across three levels during Phase One (April–June 2015). 1. School level: £1000 prize for winning school. 2. Team level: trophy for the winning team in each school. 3. Individual level: weekly prizes for highest steps and most improved within each school. During Phase Two (July–September 2015), there were individual prizes for the top three participants in each school achieving the highest average number of steps across the 14-week period. This two-phased tapered approach was designed to encourage medium-long term PA behaviour change (ie, extrinsic to intrinsically motivated PA behaviour). | ► Set graded tasks. ► Provide rewards contingent on successful behaviour. ► Provide feedback on performance. |
| Material rewards/prizes | Material rewards included coloured stickers, selfie sticks, completion certificates, cinema tickets and £10 sports vouchers. Individual prizes were awarded on a weekly basis under two categories: 'outstanding performance' and 'most improved'. | ► Prompt rewards contingent on effort or progress towards behaviour. |
| Teams | A team-based competition was developed alongside the main school competition to encourage peer support. Ten teams were created within each school (4–5 participants per team). Team captains were selected based on baseline PA data to ensure balance between teams and peer nominations to identify those 'most looked up to'. The highest placed team within each school at the end of Phase One was awarded with a trophy. | ► Plan social support/social change. ► Facilitate social comparison. ► Prompt identification as role model/position advocate. |
| Pedometers | Participants were given a Fitbit Zip pedometer and asked to wear throughout every day of the intervention (Phases One and Two). Pedometers provided participants with feedback on daily steps and were uploaded to the study website via the Fitbit App or using a wireless dongle located at designated areas within schools. | ► Goal setting (outcome). ► Prompt self-monitoring of behavioural outcome. ► Provide feedback on performance. |
| Website | Pedometer data were uploaded to the StepSmart Challenge website and participants could review their daily/weekly scores and view the competition leader board. The website included the provision of motivational messages, weekly challenges and links to other PA resources. | ► Goal setting (outcome). ► Prompt self-monitoring of behavioural outcome. ► Provide feedback on performance. |
| Workbook | A short workbook was given to participants at the start of the intervention. This included 'fun-facts', tips and challenges to promote PA behaviour as well as a section for the participant to record weekly step target (individual and team). | ► Provide information on consequences of behaviour in general. ► Goal setting (outcome). ► Prompting generalisation of a target behaviour. |

PA, physical activity.

This presented an opportunity to further understand potential mechanisms of behaviour change, and how perceptions and experiences of the intervention changed over time (preintervention, during intervention and postintervention).[21] Understanding why certain choices were made can produce more insightful and considered interpretation of behaviour change.[22] Such approaches are particularly valuable in providing a different perspective in assessing interventions or as part of process evaluations.[23]

**Focus group participants**

Baseline focus groups were conducted in each intervention school with a subsample of trial participants using a purposive sampling strategy whereby teachers identified potential participants with a range of PA levels from low-to-high as well as those with mixed educational ability. To reduce selection bias, the researchers discussed the importance of having a range of views within focus groups before participants were selected. However, it was considered that teachers were best placed to make these judgements as researchers did not know any of the participant's backgrounds and would not be aware of hidden conflicts or instances of bullying that may have influenced the group dynamic and quality of data. Those interested in taking part were given a study information sheet by the teacher explaining the purpose of the focus groups to read themselves and then give to their parent(s)/

**Table 3**  Characteristics of focus group participants

| Participant | | Team | Average steps per day (measured using Actigraph GT3X accelerometers) | | |
|---|---|---|---|---|---|
| | | | Baseline | Postintervention | 12-month follow-up |
| 1 | School C | C10 | 9949 | 8576 | No valid data* |
| 2 | School C | C6 | No valid data* | No valid data* | No valid data* |
| 3 | School C | C6 | 8815 | 13127 | No valid data* |
| 4 | School C | C7 | 9325 | 4099 | 4099 |
| 5 Winner of 'Most Improved' | School C | C1 | 9264 | 6687 | 14246 |
| 6 Winner of 'Walker of the Week' Winner of Summer Competition | School C | C5 | 13326 | 9563 | 8039 |
| 1 Winner of 'Walker of the Week' Winner of Summer Competition | School D | D2 | 10940 | 10684 | 11784 |
| 2 | School D | D9 | 2787 | No valid data* | No valid data* |
| 3 Winner of 'Most Improved' | School D | D6 | 9737 | 7160 | 7160 |
| 4 | School D | D5 | 6555 | No valid data* | 4088 |
| 5 | School D | D5 | 2782 | No valid data* | 5426 |
| 6 Winner of 'Most Improved' | School D | D7 | 9253 | No valid data* | No valid data* |
| 1 | School E | E7 | 6495 | 13080 | 6129 |
| 2 | School E | E7 | 7330 | No valid data* | 9440 |
| 3 | School E | E2 | 6583 | No valid data* | No valid data* |
| 4 | School E | E9 | 5915 | No valid data* | No valid data* |
| 5 Winner of 'Walker of the Week' Winner of Summer Competition | School E | E6 | 14153 | 13998 | 8179 |
| 6 Winner of 'Most Improved' | School E | E3 | 14113 | No valid data* | 9988 |
| 7 Winner of 'Walker of the Week' Winner of Summer Competition | School E | E3 | 11330 | No valid data* | 5909 |

*No valid data=unreturned accelerometer or no valid 3-day measurement of data.

guardian(s). Parental (or guardian) opt-out consent and participant consent was sought from all participants.[24]

The focus groups were repeated during the study, following the same participants on their journey through the trial. This provided rich contextual data to explore the views and experiences of participants over time. Data were collected 1 month preintervention (T0), at the end of the team competition (8 weeks) (T1), end of the individual competition (postintervention) (24 weeks) (T2) and at 12-month follow-up (T3). Focus groups were conducted on school premises and were audio recorded. The researchers verbally reaffirmed consent to participate at the beginning of each focus group. No other participants were present at the time of the focus groups.

Focus groups were semistructured, based on topic guides (see online supplementary material I) exploring

core concepts at each time point. The topic guide was not piloted but developed iteratively reflecting on the data gathered from the focus groups from previous time points. Thus, emerging themes were explored across time points to chart changing views, experiences and PA behaviour. During all focus groups, the researcher summarised information at the end of each section and questioned understanding as a form of participant verification.[25]

Core concepts explored included:
1. General views and experiences of the intervention and intervention components.
2. Motivation to be active and to sustain activity long term.
3. Extrinsic motivators including (A) competition (eg, Does the competition motivate you to walk?); (b) material incentives (eg, Was the opportunity to win a

prize something that motivated you?); and (C) motivation for PA (eg, What motivates you to be active?).

4. SDT concepts including (A) autonomy (eg, What new ways have you found to be active?); (B) perceived competence (eg, How did it make you feel when you compared your steps to those of the class?); and (C) relatedness (eg, Do you think friends are important in terms of how active you are?).

Focus groups were conducted by RC (male, PhD student), PB (male, postdoctoral researcher) and RO (female, postdoctoral researcher). PB and RO are experienced qualitative researchers and have facilitated focus groups with adolescents previously. RC had undergone a number of formal training courses in the facilitation of focus groups and thematic analysis methods. RC was accompanied to the focus groups by either RO or PB. Saturation of the data was discussed between PB and RC. None of the researchers had any relationship with the participants.

### Data analysis

Focus group recordings were transcribed verbatim and anonymised. Data were imported into NVivo (V.10, QSR, Southport, UK) to manage and analyse the transcripts. Analysis was undertaken using the Thematic Analysis Framework at the semantic level using a recurrent cross-sectional approach.[26] Initially researchers (RC and PB) familiarised themselves with the data. A sample coding frame was developed by the researchers independently and refined iteratively with subsequent discussions. As a result, three coding frameworks were generated, one for each core theme. Illustrative quotes supporting emerging

**Table 4** Overview of the number of participants in (and duration of (in minutes)) each focus group at each time point

| Intervention schools | Time points of each focus group | | | |
| | Baseline (T0) | 8 weeks (T1) | 24 weeks (T2) | 52 weeks (T3) |
| --- | --- | --- | --- | --- |
| School C (all boys) | 6 (35) | 6 (21) | 5 (38) | 5 (35) |
| School D (all girls) | 6 (34) | 5 (37) | 6 (40) | 2 (31) |
| School E (all girls) | 7 (36) | 7 (41) | 6 (24) | 7 (24) |

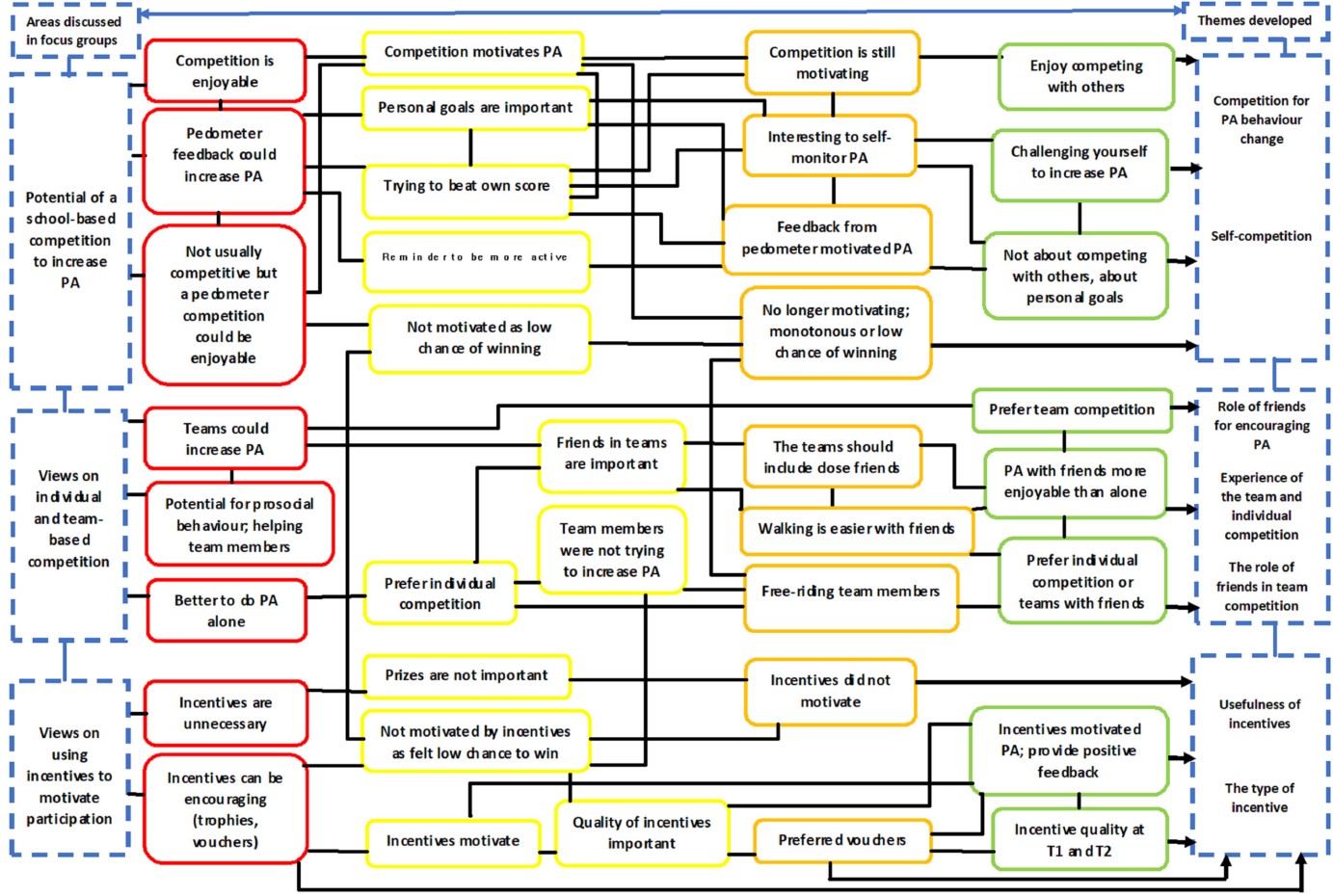

Time points: T0 = Red ; T1 = Yellow; T2 = Orange; T3 = Green

**Figure 1** Diagrammatic representation of the temporal thematic pathways that developed during focus group interviews. PA, physical activity.

themes were highlighted and agreed by researchers. Transcripts were not returned to participants for comment, and they did not provided feedback on findings.

The dataset was systematically coded using an inductive approach and codes were generated to give a summary of elements of analytic interest. Once coded, researchers identified potential themes from clusters of codes of similar meaning as well as patterns of responses across codes. Three central themes were identified at T0 and developed at subsequent time points. The coding frame was then discussed with ME and further refined. RC and ME then met multiple times to discuss and refine codes until a final coding frame was applied to all data. These themes consisted of (1) incentives; (2) competition; and (3) influence of friends on PA. The temporal changes in the views and experiences of participants and the influence of these components on the process of behaviour change were explored under each theme. Researchers (RC and ME) analysed the data together to further refine subthemes to ensure assertions were accurately reflected. Codes were not validated with study participants due to the time commitment that they had already provided due to the qualitative longitudinal design. However, given the nature of the QLR design, core concepts that were apparent at T0 were revisited at subsequent time points to test the validity of the theory.

## RESULTS

Table 3 details the characteristics of participants in the focus groups and demonstrates that the purposive sampling strategy was successful in recruiting participants of mixed gender, socio-economic status (SES), from different teams, those who won prizes and those who did not win, and PA levels. Table 3 displays a breakdown of the focus groups conducted. Twelve focus groups were conducted (mean duration 33 min; range 21–41 min (table 4)). Focus group participants present at each time point varied due to participant absences (mean six participants; range 2–7 participants). No participants refused to participate or dropped out.

The following results section details the themes and subthemes identified from the thematic analysis. This includes a diagrammatic representation (see figure 1) of how themes continued to evolve as new data emerged at each time point.

### Theme 1: competition
Three subthemes emerged under this theme: (A) usefulness of competition for PA behaviour change; (B) self-competition for PA behaviour change; and (C) experiences of the team and individual competition.

### Subtheme A: usefulness of competition for PA behaviour change
At T0, participants suggested the competition would motivate them to work harder, and it was generally viewed as a motivating factor to become more physically active:

[T]hat's what motivates me. (School C, male 5, T0).

[T]hat's what pushes people on. (School C, male 6, T0).

[I]f I was actually in a competition I'd actually walk everywhere. (School D, female 6, T0).

As interviews progressed (across time points), these early conceptualisations were developed further. For example, it was clear from T1 onwards that for a subsample of participants in all schools, the desire to compete was not sufficient; the goal of winning was paramount:

[W]e like the winning but we don't like the losing. (School E, female 3, T0).

[T]o try and win like every week after that. (School C, male 6, T3).

[Y]eah, because you just wanted to win. (School D, female 6, T3).

Yet while this subsample was extremely motivated during Phase One, when the competitive elements (against others) lessened during Phase Two, the intervention became monotonous or 'boring':

[I] think that's boring you know, who would want to know how many steps you're taking? (School D, female 2, T2).

[A]t the start like, like you were quite motivated and then it just got more on and then you just forgot to wear it some days and it just got quite boring. (School C, male 4, T3).

[I]t was just like the same thing every day. (School C, male 4, T3).

For others, the benefits of competition related to their perceived chances of winning. If this was believed to be remote then motivation lessened. This became clear at T1 as the researcher reflected on earlier (T0) responses given by participants.

[W]henever you found out that you're actually losing there's just no point. (School D, female 2, T1).

[I]t's just sort of cause you knew you probably weren't going to win so you're just like there's really no point in wearing it [pedometer]. (School D, female 6, T2).

[W]ell I just really gave up whenever X just won everything. I really did. I just stopped. (School D, female 3, T3).

### Subtheme B: perceptions of the usefulness of self-competition for PA behaviour change
The potential use of the pedometer for self-directed goals or 'self-competition' (competing against oneself) was considered promising at T0:

[I]f its showing you how many steps you're taking then you could challenge yourself to take more every day. So if you took 2000 steps 1 day you could try like try take more the next day. So it's like challenging yourself. (School D, female 1, T0).

Self-competition was shown to be a motivating factor throughout the intervention for most participants. One participant (school D) viewed 'getting better' and improving step counts as meaningful, reducing the negative effect of not winning prizes. This aligns closely with SDT and shows possible progression towards intrinsically based motivation for PA.

> [T]rying to beat your day before target. (School C, male 5, T1).

> [I] loved to see like how many steps you were actually taking like when you're beating your scores as well like you check it one day and then the next day your gonna try and beat it. (School E, female 1, T2).

> [N]o, it's alright because I was getting more each day so I was getting better; so it was alright. (School D, female 6, T3).

### Subtheme C: experiences of the team and individual competition

The intervention incorporated two formats of a pedometer competition: a team-based competition (8 weeks duration) and an individual competition (14 weeks duration). During the individual competition, participants competed against others from their school. At T0, the team-based competitions were seen to have the potential to better encourage PA:

> [Y]ou can work together as a team. (School D, female 4, T0).

> [I]f you were in like a group with more active people like you'd be sad that you're not as good as them but it would kind of push you to be as good as them. (School E, female 7, T0).

When asked to compare the individual with the team-based competition at T2, many participants from school C (all male) were more motivated by the team-based competition. This was due to the support provided by the team or peer pressure from not wanting to let '*your team down*':

> [I]'d probably say it's easier with the support rather than individually. (School C, male 6, T2).

> [Y]ou didn't want to let your team down. (School C, male 6, T2).

Reflecting on the intervention at T3, there were clear differences regarding experiences of the competition. School C (all boys) participants continued to feel positive about the team competition.

> [T]he team's a lot better. Like its more… you are just like together. (School C, male 1, T3).

> [T]he individual is quite boring. (School C, male 4, T3).

The girls' schools (schools D and E) tended to favour the individual competition, as they did not have to rely on their team members, or for logistical reasons, such as the inability to meet up and 'organise stuff'.

> [Y]ou don't have to depend on anyone else. (School D, female 3, T3).

> {Y]ou know like sometimes you don't live close to your friends so you can't always organise stuff, which is a problem. So I think the individual one. (School E, female 7, T3).

One disadvantage of the team competition was the issue of free-riding team members (ie, a member of a team that obtains benefits from membership but is not seen to contribute a fair share of the work needed to accrue the benefit).[27] In school D (all girls), free-riding was evident.

> [W]ell in the group you didn't really have to do anything cause the rest of them could do it but like by yourself like I don't know you just lose it altogether cause you don't walk. (School D, female 2, T2).

> [W]e didn't really have to worry about it cause like the rest of them would've like walked anyway. (School D, female 2, T2).

## Theme 2: incentives

Two subthemes were identified including (A) type of incentive and (B) perceptions of usefulness on incentives.

### Subtheme A: type of Incentive

The provision of material incentives in this study was contingent on doing well in the competition, rather than being contingent on PA behaviour change. The types of incentives suggested and discussed by participants included recognition-based incentives (eg, trophies) and material incentives (eg, vouchers). Males tended to favour recognition-based incentives, whereas material incentives with a higher monetary value were largely proposed by females. While this was apparent at T0, the QLR approach enabled the researchers to revisit this at subsequent time points to test the validity of the theory. When asked what type of prizes they would like, male participants suggested:

> [A] medal or a trophy. (School C, male 5, T0).

> [R]ugby ball. (School C, male 4, T0).

In contrast, females often suggested the use of material incentives.

> [V]ouchers for clothes. (School D, female 6, T0).

> [T]opshop (clothes store) vouchers . (School D, female 1, T0).

> [M]oney. (School E, female 7, T0).

### Subtheme B: perceptions of usefulness of incentives

At T1 and T2, many participants viewed incentives favourably. Participants suggested that the incentives were desirable and encouraged them more during the intervention.

> [E]very week cos you know it's like running out of time for like the prizes, just really want to get one. (School C, male 6, T1).

[T]hat they weren't just like wee rubbish prizes they were really good ones. (School E, female 4, T2).

At T3, when reflecting on the intervention, participants still viewed the incentives as a motivating factor as they were 'good' prizes and provided acknowledgement for achievement:

[Y]eah, they look good. Like the prizes were really good. (School C, male 4, T3).

[Y]eah, I think [it] was good actually. Just to keep people motivated. (School E, female 7, T3).

[L]ike you know you are being acknowledged, like when you get prizes. (School E, female 3, T3).

Some participants did not win any prizes over the course of the intervention. When these participants discussed the instances when their peers won prizes, there was a clear sense of disappointment, with a number stating that it was '*annoying*':

[K]inda annoyed you when people like brought out their ten pound of cinema tickets and yeah it's like kinda annoying. (School C, male 3, T2).

[L]ike it annoyed me that I didn't get one. (School E, female 6, T2).

[I]t just made me sad. (School E, female 2, T2).

### Theme 3: influence of friends
Two subthemes emerged, including: (A) the role of friends in general for encouraging PA behaviour and (B) the role of friends in team competition.

#### Subtheme A: role of friends in general for encouraging PA behaviour
At T0, participants suggested that PA was more enjoyable with friends, and the social support provided by friends encouraged participation in PA:

[I]t's about encouraging each other to do stuff. (School C, male 5, T0).

[I]f they want to go for a run you will want to go for a run with them. (School C, male 4, T0).

[Y]eah, because you want to be doing it with them so you can enjoy yourself. (School C, male 6, T3)

[G]ood friends will help you yeah. (School D, female 6, T0).

[X] only lives up the street so we go for runs most days after school. (School E, female 3, T0).

Participants in school E discussed the continued positive influence of friends on PA motivation at T1 and T2. This added additional depth to T0 findings by showing friends as providing a social acceptable context in which to be active. Feeling 'scundered' (colloquialism for embarrassed) when walking alone is offset when provided the social support of friends:

[M]ake you feel like I'm going to be scundered [embarrassed] walking about alone but when you have

your friend with you like you'd be more encouraged to do more walking if you're like walking with your friend. (School E, female 2, T2).

[H]ardly just like go a walk about yourself about the street like a big loner. (School E, female 4, T2).

#### Subtheme B: role of friends in team competition
Participants suggested that the influence of friends and a sense of connectedness was necessary within teams in order for them to work together and be competitive. Although some participants in school C felt it would be '*good to have at least one friend or two*' (School C, male 6, T3), they did not want to pick their own teams as they believed this might produce imbalanced teams with the more physically active individuals going into the same team. For these participants, the combination of friends within a team and homogeneity between teams was important:

[N]o cause then they could get really unfair. (School C, male 1, T2).

[C]ause all the active people could go in one team and then the inactive so it wouldn't work out. (School C, male 1, T2).

In contrast, participants in schools D and E wanted the opportunity to choose team members, preferring to be in teams composed of their friends:

[I] wanted to choose my own team. (School D, female 4, T1)

[N]o I think it should just be like your own group like friends like five of each of them. (School E, female 3, T1).

[B]ecause like [if] you don't like people in your team you're just going to be like 'nah not even going to talk to you'. (School E, female 6, T2).

Figure 1 illustrates the various thematic pathways that developed during focus group interviews at T0 (red), T1 (yellow), T2 (orange) and T3 (green). Taking the first theme (competition) as an example, the researchers considered the emergence of two distinct groups at T0. These were (1) *physically active participants* who viewed a pedometer competition as a means of further increasing their active lifestyle and (2) *less active participants* who viewed the pedometer competition as an opportunity to become more active.

At T0, physically active participants were particularly 'excited' about a pedometer-based competition, perceiving it as an enjoyable process. Some of these participants appeared to engage regularly in competitions. The less active group were more cautious but felt a pedometer-based competition might provide an acceptable context through which PA may be enjoyable. As such, these early ideas/themes were represented at T0 (red).

As interviews progressed through T1 and T2, the research team observed changes in relation to participant's views of competition (both in a general sense as

well as relating directly to the StepSmart Challenge). Moreover, the QLR approach enabled the research team to frame these changes within the context of data revealed at baseline (T0). For example, physically active members who regularly engaged in competitions (outside of the StepSmart Challenge) at T0 continued to enjoy the intervention at T1 and T2. However, for a subset of this group (where winning was more important than competing), motivation lessened at T2. Figure 1 also illustrates that for participants who were less active, motivation and engagement decreased much sooner (at T1) and continued to decrease into T2 and T3. In some cases, these participants appeared to have been motivated solely by the material incentives, thus the perceived failure of 'not winning a prize' was interpreted as negative feedback and reinforced negative schemas around PA.

SDT maintains that an activity that is stimulating is an important aspect of sustained motivation. Throughout the intervention, the importance of self-monitoring and the importance of personal goals was prominent for all participants. The concept of self-competition provides an opportunity to challenge oneself and can be supportive of feelings of competence. Self-competition provided an opportunity for all participants to receive positive feedback by meeting the goals they set for themselves and could lessen the impact of not winning prizes. By adopting a QLR approach, themes generated in earlier focus groups evolved and could be tested as new data emerged. The same depth would have been difficult to achieve within a pre-test and post-test design.

## DISCUSSION

Participant's generally had positive experiences and views of this gamified PA intervention. Results suggested that the gamified design may have had an important role to play in encouraging adolescents to engage in PA, and in creating interventions that are fun and enjoyable. The findings also suggested that core concepts of SDT are compatible with gamified BCTs and that some game elements may require tailoring for specific populations, for example, different genders.

In general, the use of a gamified pedometer competition was viewed favourably by participants. The goal of winning was very important for some and was key to sustaining their motivation to be active. This could be linked to the provision of material incentives, which was contingent on 'winning' the competition. Over the course of the intervention, material incentives continued to motivate some participants. A possible mechanism could be that positive feedback provided by winning prizes and doing well in the competition helped develop an individual's intrinsic motivation by improving feelings of competence.[28] The positive effect of material incentives for health behaviour change with children and adolescents has also been shown in previous studies.[29–32]

The long-term effect is less clear with some studies showing that positive effects dissipate over time.[33] This could be due to habituation to the extrinsic motivators being offered[34] or a 'crowding out effect' of intrinsic motivation[35] once extrinsic incentives are removed. However, to date, this hypothesis has not been tested or supported in 'real world' interventions.[28 36–38]

Other participants felt demotivated from the outset as they believed they had no chance of winning. Some became less enthusiastic about the competition if other participants consistently had a greater number of steps and were disappointed at not winning a prize. Previous work has suggested that competition can affect participants' self-evaluation of their competence to perform the task.[39 40] If a participant loses, and their loss is attributed to low ability, this can negatively impact behaviour.[41] Therefore, participants may choose not to compete or not engage in the competition with maximum effort.[42] This helps to provide some explanation for the loss, other than low ability, thus preserving the participant's self-esteem and self-efficacy.

Some participants indicated that they became gradually less interested because of the repetitive nature of the pedometer competition. These findings are supported by a large body of literature that suggests that extrinsic motivators can have a short-term positive effect on motivation that is not maintained.[33 36 37 43] Extrinsic motivators such as competition and material incentives could be used to initially stimulate the interest of participants, especially those with lower levels of PA.[44] However, a key learning point would be to transition to more intrinsically motivating forms of PA, and thus, the incorporation of BCTs that focus on these behaviours would be useful.[45 46]

The competition had various levels: rewards could be offered to the highest achieving team, the highest achieving individual or to anyone on the basis of achieving some personal goals (self incentive). The findings showed distinct perceptions regarding the value of each. For example, males tended to prefer the team competition and suggested they would try harder to contribute to the team, and found the team environment supportive and enjoyable. Maculada[47] suggested that males find team affiliation important and a way to be accepted by peers and to feel a sense of belonging with the group. Team-based PA interventions have been shown to be effective[10 11] and may be less harmful than individual competitions.[48] Conversely, females favoured individual competition; how well a participant did in the competition was not dependent on the effort of others, mitigating to the problem of free-riding.[49] One solution may be to distribute incentives equitably (ie, proportionate to effort and contribution) to team members rather than distributing them equally,[50] thereby reducing free-riding and increasing effort.[51 52]

Self-competition[53] was seen as a prominent positive influence of PA. Participants often used the pedometers for feedback, to self-monitor and set personal step goals. Creating achievable personal goals may also play a part in mitigating the potential negative effects of extrinsic motivators by emphasising competence (by meeting goals and receiving positive feedback), autonomy (as participants

are free to choose which activities they pursue to increase step counts) and maintaining self-efficacy. Self-competition with the use of intrinsic goals was enjoyed by all participants in the focus groups, regardless of success in the overall competition. Self-competition allowed participants to be autonomous and to create achievable challenges such as walking more steps than during the previous day. Therefore, self-competition could be a way to develop autonomous identified or integrated regulation, which has been shown to have benefits for PA motivation.[54 55] Autonomy-supportive elements such as self-competition could consequently stimulate the development of habit formation.[56]

The significance of friends for influencing PA behaviour has also been frequently cited in the literature.[57–59] The participants' feelings on team composition and the influence of friends reinforce the psychological need for relatedness, a core construct of SDT. Participants from all schools felt that a sense of connectedness to the group was important for an effective team competition. Other research shows that adolescents value opportunities for social interaction,[39] and so team membership could have a positive effect on PA motivation. Participants stated that friends provided support, encouragement and help with the enjoyment of PA.

### Reflections on the QLR approach

The authors acknowledge the difficulty in mapping temporal changes, especially in focus groups, where there may not be sufficient time or opportunity to explore individual's views in detail. Nonetheless, figure 1 is a simplified, but useful, thematic illustration of general (group-level) consensus over a 1-year period.

The complexities contained within each pathway highlight the difficulty in developing a group-based PA intervention that will motivate all participants in a similar manner. It also illustrates the interrelated nature of the themes and how experiences of one aspect of the intervention can influence other components. However, recognition-based incentives, the provision of feedback on performance and opportunities for social connectedness were shown to be key gamification strategies with potential for motivating PA throughout the intervention period. This is in line with SDT, which posits that supporting innate desires, competence and a sense of relatedness with others could help achieve a higher quality of motivation that is long lasting.

### Strengths and limitations

A major strength of this study was the novel use of a longitudinal design,[21 60] using the same participants in repeated focus groups at baseline, post-intervention and 1-year follow-up. This enabled the study of how participants' views, experiences and PA behaviour evolved over time. The findings are robust as assumptions, views and experiences can be tested and retested in subsequent sessions, and researchers build relationships with participants due to the repeated exposure that can encourage

disclosure. In addition, having a number of researchers involved in the data collection and analyses reduces selectivity and researcher bias.

A focus group method was chosen as it provided an opportunity for the group to discuss issues among themselves and reach consensus, gathering multiple viewpoints and representing 'everyday' conversation. However, the approach has been criticised for lacking depth, particularly when conducted with young people as they tend not to elaborate on discussion points. It may also have been useful to combine this approach with 1:1 interviews to reduce peer pressure and ensure coherence of responses at different schools. Only three focus groups participated in the study, and all were single-sex schools. Consequently, the purported gender differences may be an artefact of differences in SES as well as or in addition to gender differences. Finally, there was good retention of participants in the qualitative longitudinal design, with the exception of T3 in which four (out of six) pupils were missing from school D owing to a timetable clash that was beyond the control of the research team.

### Conclusions

Preferences for gamified elements including team or individual competitions and the influence of friends on PA behaviour were highlighted. The use of a longitudinal qualitative design enabled exploration of temporal changes in participants' views and experiences, and exploration of potential mechanisms of behaviour change. This study suggests that the three core constructs for self-motivation in SDT could be important factors for motivating PA in adolescents via competition and the use of material rewards delivered through gamification. This supports previous research that proposes benefits in providing opportunities for autonomy, perceived competence and relatedness.[61]

**Correction notice** This article has been corrected since it first published. The Open access licence has been changed to a CC BY licence.

**Acknowledgements** The authors wish to thank the pupils and teachers who were involved in the development of this intervention and those who participated in the study. The authors wish to acknowledge funding from a HSC R&D (NI) Enabling Research Award.

**Contributors** RFH had the initial idea for the study and led the writing of the grant with significant contribution from MAT and FK. RFH, PB, MAT and FK were involved in the design of the study. RC, PB and RO facilitated the focus group discussions. RC, PB, RO and ME were involved in the analysis of the data. All authors were involved in the interpretation of the data. All authors provided substantial comments on the drafts of the manuscript and approved the final version.

**Funding** The authors received funding from a HSC R&D (NI) Enabling Research Award. RFH is supported by a NIHR Career Development Fellowship and acknowledges funding support from the HSC Research and Development Division. The work was undertaken under the auspices of the UKCRC Centre of Excellence for Public Health Research Northern Ireland, which is funded by the British Heart Foundation, Cancer Research UK, Economic and Social Research Council, Medical Research Council, the National Institute for Health Research and the Wellcome Trust.

**Disclaimer** The funders had no role in study design, data collection, data analysis, data interpretation, or writing of the report.

**Competing interests** None declared.

**Patient consent** Not required.

**Ethics approval** School of Medicine, Dentistry and Biomedical Science Research Ethics Committee (Queen's University, Belfast) (Ref: 15.09).

**Provenance and peer review** Not commissioned; externally peer reviewed.

**Data sharing statement** No additional data are available.

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
