## [Reviewer comments · BMJ Open]

ARTICLE DETAILS

TITLE (PROVISIONAL)	Exploring the use of a gamified intervention for encouraging physical activity in adolescents: A qualitative longitudinal study in Northern Ireland
AUTHORS	Corepal, Rekesh; Best, Paul; O'Neill, Roisin; Tully, Mark; Edwards, Mark; Jago, Russ; Miller, Sarah; Kee, Frank; Hunter, Ruth

VERSION 1 – REVIEW

REVIEWER	Marilyn Lennon Digital Health and Wellness Research Group University of Strathclyde Glasgow, UK.
REVIEW RETURNED	14-Nov-2017

GENERAL COMMENTS	The results from this study are not surprising – and are not novel. There are many tens of papers now in Human computer interaction, physical activity and health journal and conferences that all have identified these key findings already. The authors themselves cite papers that state upfront that competition, incentives, and peer support are all factors that have been identified in gamification as having an effect. Are the authors merely trying to confirm this body of work? If so I am not sure of the size or value of the contribution of the paper. What have the authors added that is new? The methods used are appropriate (focus groups with same people over time during an intervention) but are somewhat limited and there are a few issues. - I appreciate that the focus groups might have been exploratory – but I really expected a clear research question or set of research questions to be stated somewhere near the start of the paper – to allow me as reader to understand what was being investigated in the focus groups better. The authors say “to explore core concepts” or “to explore experiences” a lot and do not spell out to the reader clearly enough what was being investigated in these focus groups from the outset. The outline of page 8 is good – but should be more explicitly formed as research questions or aims at the start of the paper.- Small numbers and so the results are limited in their generalisability. Sampling – why did teachers select students for the focus groups – this would introduce a huge bias – teachers always pick pupils that will represent the school well etc. so I just can't understand why the research team were not allowed to select participants to reduce the bias here. Why were only treatment group involved – would it have made any sense to have opinions from the control groups as well? - Why were no field notes taken? Seems strange to mention
---

	this and not justify it briefly. - - why were codes not validated with participants? This is common practice and acceptable but might be worth rationalising since you mentioned it. One of the key potential strengths/is the additionality of a temporal analysis. The authors promise: “qualitative charting of the perceived behaviour change process over a one-year period,..... to further understand potential mechanisms of behaviour change, and how perceptions and experiences of the intervention changed over time (pre, during and post intervention). I think the authors need to make more of this throughout. Some sort of diagrammatic summary of how (and if) certain codes/themes changed over time would have helped a lot. “Understanding why certain choices were made can produce more insightful and considered interpretation of behaviour change.” Making more of this in the discussion would add to the potential contribution of the paper (especially given that the actual themes themselves are not new). In Table 2 – it would make a lot of sense to acknowledge where the BCTs techniques chosen ‘came from’ – link back to the literature or the design choice/rationale for each of these. I appreciated these being summarised – but these additionally details would add strength and value to the choices made and I justifying that they were evidence based and not just plucked out of thin air or even cherry picked. The topic guide is very useful in helping the reader to appreciate that the questioning in the focus group was well thought and directed around the research questions – but these research questions need to be make more explicit earlier in the paper. I appreciated the COREQ checklist but the authors don’t really reflect on it much (or at least don’t reference it directly if they did) throughout the paper. Overall I found the methods acceptable (with the small caveat that I had some small issues detailed above about missing details and small sample size). The study is well motivated – but the actual novelty and the overall contribution are limited in the current presentation. I feel that the authors could work on making more of the temporal analysis – and add something more critically reflective and possibly some sort of diagram to bring this aspect out more – as it is the three key themes on their own seem rather underwhelming and not new – but the longitudinal temporal analysis – of brought out more could be a valuable contribution.
--	--

REVIEWER	Aigul Mavletova Senior Research Fellow, Laboratory for Comparative Social Research; Associate Professor, Faculty of Social Sciences, National Research University Higher School of Economics, Moscow, Russia
REVIEW RETURNED	05-Dec-2017

GENERAL COMMENTS	The manuscript explores the views and experiences of adolescents who participated in a gamified 24-week primarily school-based physical activity (PA) intervention (the StepSmart Challenge) in three schools in Belfast, Northern Ireland. The study is based on the
---

analysis of focus groups. The main strength researchers report is that the study is based on a longitudinal qualitative study at four time-points (baseline, end of each of two intervention phases and 1 year follow-up). I think the study is relevant and will be interesting for a good number of researchers from different fields whose focus is on applying gamification among children and adolescents. Though I think the manuscript is good, I still have some comments:

- 1) The term gamification is not defined. Though there is a large number of definitions one can find, the authors should include the definition they were using while designing BCT gamified elements for the StepSmart Challenge. What did they have in mind while designing it this way? What BCT gamified elements they found crucial for a successful experiment? They should clarify that. It would be helpful if the authors would state gamified elements they find important for applying gamification in health-related studies among adolescents (in the "Background" section), and add a column in Table 2 with these elements in the following section.
- 2) There is a discussion in gamification field about the role of intrinsic and extrinsic motivation. Gamification is about motivation anyway. But it is not clear what the authors expected in terms of intrinsic and extrinsic motivation and how they would be related in their study. They discuss intrinsic and extrinsic motivation in the "Discussion" part, but I was wondering how they tried to start with the extrinsically motivated PA behavior and change it to intrinsically motivated behavior, and if the gamified design of the StepSmart Challenge was successful with that.
- 3) The authors should state the research problem more clearly. Now there is only a descriptive analysis of focus groups at different time points, but it is not focused on the research problem. It would make sense if the authors would pose some research questions like "why gamification in PA intervention does not work as expected among some groups of adolescents" or "how gamification can be improved while applying to adolescents in health studies" or "how gamification can be helpful for those with low PA" or "how the long-term gamification effect can be maintained" or something else. A longitudinal qualitative study can help us better understand these issues which we can find in a quantitative study. Here we can explain or make suggestions based on the analysis of focus groups. A good number of experimental studies in different fields show that gamification can be efficient among both adults and children/adolescents. However the effect size varies a lot, and the stability of this effect varies in time (the effect size within the course of time), or we can not disentangle the effect of different gamified elements. I believe this qualitative study can help us answer some of the questions related to the duration of the gamification effect and the factors that have an effect on that duration, or effects of different gamified elements on different groups of adolescents, especially on those with a low level of PA. Again, there should be more problem-focused analysis of the materials the authors have. If researchers analyze temporal changes, then for instance they can report which gamified elements have the long-term effect in terms of motivation, which of them varied most, etc. With that being said, I would ask the authors to include some basic quantitative results in their study to what they report on p.6 (l.16-19), e.g. the effect size at different time points compared to the control condition + any other relevant results depending on the research problem they would focus on.
- 4) The authors analyze data thematically. They identified three core themes: competition; incentives; influence of friends. Though it makes sense, it is still confusing to have these particular themes without any explanation or theoretical foundation. I would suggest to

	have theoretically-driven themes according to the main research problem (see point 3 above). It might be the elements of SDT theory and the indicators of these elements in their study. 5) In “Discussion” part the authors report some interesting findings but sometimes no explanation or hypotheses based on their analysis (e.g., long-term effect, lower interest – the balance between extrinsic and intrinsic motivation, etc.). 6) This part was not clear: “This could be linked to the provision of material incentives which was contingent on ‘winning’ the competition. Over the course of the intervention, material incentives continued to motivate some participants, suggesting that intrinsic motivation and autonomy were not undermined.” - How you can make such conclusions? 7) When I was reading about the difference in individual and team competitions among girls and boys I was wondering if it is an overgeneralization based on the data they have (2 “all girls” school and 1 “all boys” school”). Again if this in line with the quantitative results they have, they should report that. Some minor issues: 8) The researchers have good numbers of those who participated in focus groups at each time point except T3 with having a problem with a particular school (school D). I think one should include that fact in the “limitations” section. 9) Some components from the Table 2 are not analyzed or discussed for some reason (e.g, website) – these components did not have any effect in the study?
--	---

VERSION 1 – AUTHOR RESPONSE

Response to Review

Thank you for the comments provided by the editor and the reviewers. We have considered each of the comments in turn below and changes to the manuscript are highlighted in red text. We feel that the manuscript has greatly improved in light of these comments.

Editorial Requirements:

1. **Please revise the COREQ checklist to include specific page numbers for each item.**

Author response:

The COREQ checklist has been revised to include the specific page numbers for each item.

Reviewer(s)' Comments to Author:

Reviewer: 1

2. **The results from this study are not surprising – and are not novel. There are many tens of papers now in Human computer interaction, physical activity and health journal and conferences that all have identified these key findings already. The authors themselves cite papers that state upfront that competition, incentives, and peer support are all factors that have been identified in gamification as having an effect. Are the authors merely trying to confirm this body of work? If so I am not sure of the size or value of the contribution of the paper. What have the authors added that is new?**

Author response:

We believe that one of the main contributions and strengths of this paper is the assessment of how the views, experiences and behaviours of those participating in gamification-based interventions change over time, through novel use of a qualitative longitudinal design. To our knowledge, this is the first time that this study design has been used to explore how the

views, experiences and behaviours of those participating in gamification-based interventions change over time. We have included an additional section in the Discussion to reflect on the qualitative longitudinal approach (page 19).

Further, we have developed Figure 1 which provides a thematic illustration of general (group-level) consensus and thematic stability over a 1 year period. The complexities contained within each pathway highlight the difficulty in developing a group-based PA intervention that will motivate all participants in a similar manner. It also illustrates the interrelated nature of the themes, and how experiences of one aspect of the intervention can influence other components. However, recognition-based incentives, the provision of feedback on performance, and opportunities for social connectivity were shown to be key gamification strategies with potential for motivating PA throughout the intervention. This is in line with Self Determination Theory (SDT) which posits that supporting the innate desires competence and a sense of relatedness with others, could help achieve a higher quality of motivation that is long-lasting.

3. The methods used are appropriate (focus groups with same people over time during an intervention) but are somewhat limited and there are a few issues.

Author response:

The authors have acknowledged methodological issues within the limitations section (page 20). However, as the reviewer has noted we feel the temporal analysis is a particular strength that enhances the quality of the data produced. To our knowledge, this is the first time that this study design has been used to explore how the views, experiences and behaviours of those participating in gamification-based interventions change over time.

4. I appreciate that the focus groups might have been exploratory – but I really expected a clear research question or set of research questions to be stated somewhere near the start of the paper – to allow me as a reader to understand what was being investigated in the focus groups better. The authors say “to explore core concepts” or “to explore experiences” a lot and do not spell out to the reader clearly enough what was being investigated in these focus groups from the outset. The outline of page 8 is good – but should be more explicitly formed as research questions or aims at the start of the paper.

Author response:

Thank you for this comment. We had formed specific apriori research questions before beginning the study. These have now been added as study objectives within the introduction (page 4-5) and are detailed below. We believe that such questions are entirely consonant with guidelines for process evaluation (Moore et al, 2015).

Reference: Moore, G.F., Audrey, S., Barker, M., Bond, L., Bonell, C., Hardeman, W., Moore, L., O’Cathain, A., Tinati, T., Wight, D. and Baird, J. 2015. Process evaluation of complex interventions: Medical Research Council guidance. *BMJ*. **350**, ph1258.

Study objectives included:

1. To explore key aspects of a gamified PA intervention over a 1-year period using a Qualitative Longitudinal Research method.
2. To discuss key issues relating to the intervention, such as PA opportunities/barriers, the value of competition and types of rewards etc.
3. To explore the key influences of PA and to determine who benefitted from the intervention, how and why it worked for them;
4. To qualitatively chart changes in behaviours, opinions or views as a result of participating in the intervention.

5. Small numbers and so the results are limited in their generalisability.

Author response:

The optimum sample size for a qualitative study is driven by rather different considerations to those that dictate a quantitative analysis. Data saturation and sample diversity are often principle concerns that support a qualitative study's explanatory coherence. In that sense, generalisability is not a particular pre-requisite for qualitative research (Leung, 2015). Our focus is on internal validity, which we feel is a strength given the longitudinal approach applied.

Reference: Leung L. Validity, reliability, and generalizability in qualitative research. *J Family Med Prim Care*. 2015; 4(3): 324–327. doi: 10.4103/2249-4863.161306

6. Sampling – why did teachers select students for the focus groups – this would introduce a huge bias – teachers always pick pupils that will represent the school well etc. so I just can't understand why the research team were not allowed to select participants to reduce the bias here.

Author response:

We have added the following details to the manuscript to better explain our sampling approach for the focus groups (page 6-7). Primarily teachers were asked to assist with the selection of participants to account for any unseen group conflict or bullying. We wanted all participants to feel comfortable in sharing their views and this was an important ethical consideration for us. We explained to teachers (pre-baseline) the importance of having a mix of opinions in the groups (e.g. pupils with a range of physical activity levels and for mixed educational ability), and both the pupils and teachers knew that the purpose of the focus group discussion was not to represent the school (well or otherwise) but to offer views on the intervention. The demographic characteristics of the focus group participants (see Table 3) suggests that this was achieved.

7. Why were only treatment group involved – would it have made any sense to have opinions from the control groups as well?

Author response:

The aim of this study was to explore how views, opinions and behaviours changed when taking part in a gamified intervention (see page 4-5 for specific study objectives). Therefore, it would not have been appropriate to include participants from the control group.

8. Why were no field notes taken? Seems strange to mention this and not justify it briefly.

Author response:

All interviews were audio-recorded and transcribed as per standard practice. The reference to field notes was included as per a requirement of the COREQ checklist. We have removed this statement from the manuscript to save causing confusion but included it in the COREQ checklist supplement.

9. Why were codes not validated with participants? This is common practice and acceptable but might be worth rationalising since you mentioned it.

Author response:

During all focus groups, the researcher summarised information at the end of each section and questioned the pupils' understanding as a form of participant verification. Codes were generated, validated and checked for accuracy jointly by the first author (RC, a PhD student) and an experienced qualitative researcher (ME).

Codes were not validated with study participants because they had already made a substantial commitment to the study due to the qualitative longitudinal design. However, given the nature of the QLR design core concepts that were apparent at T0 were revisited at subsequent time points to test the validity of the theory. This justification has been added to the manuscript (page 8-9).

10. One of the key potential strengths/is the additionality of a temporal analysis.

The authors promise:

“qualitative charting of the perceived behaviour change process over a one-year period,..... to further understand potential mechanisms of behaviour change, and how perceptions and experiences of the intervention changed over time (pre, during and post intervention).

I think the authors need to make more of this throughout. Some sort of diagrammatic summary of how (and if) certain codes/themes changed over time would have helped a lot.

“Understanding why certain choices were made can produce more insightful and considered interpretation of behaviour change.”

Making more of this in the discussion would add to the potential contribution of the paper (especially given that the actual themes themselves are not new).

Author response:

Thank you for these useful suggestions. The manuscript now contains a diagrammatic summary (figure 1) of theme development across the various time points. The authors have also inserted a narrative summary of this figure (page 16-17) and inserted a new section within the discussion (page 19).

11. In Table 2 – it would make a lot of sense to acknowledge where the BCTs techniques chosen ‘came from’ – link back to the literature or the design choice/rationale for each of these. I appreciated these being summarised – but these additionally details would add strength and value to the choices made and I justifying that they were evidence based and not just plucked out of thin air or even cherry picked.

Author response:

Thank you for this comment. The BCT’s are taken from Michie et al (2013) and we have now inserted this reference into the table. A more in-depth explanation regarding BCTs is available within the main feasibility paper, which is currently under review with another journal. This manuscript can be provided to the reviewers on request if required. Briefly, these BCTs were primarily informed by self-determination theory (Ryan and Deci, 2002) and self-regulation control theory framework (Carver and Scheier, 1982: 1998). Motivational messages (persuasion) and social support (vicarious experience) were also introduced to promote self-efficacy perceptions (Bandura, 1986).

Michie S, Richardson M, Johnston M, et al. The behavior change technique taxonomy (v1) of 93 hierarchically clustered techniques: building an international consensus for the reporting of behavior change interventions. *Ann Behav Med* 2013;46(1):81-95.

12. The topic guide is very useful in helping the reader to appreciate that the questioning in the focus group was well thought and directed around the research questions – but these research questions need to be make more explicit earlier in the paper.

Author response:

Thank you for this helpful comment. The authors have now included research aims and objectives in the introduction (page 4-5) in order to make this clearer.

- 13. I appreciated the COREQ checklist but the authors don't really reflect on it much (or at least don't reference it directly if they did) throughout the paper.**

Author response:

The items in the COREQ checklist are all reflected in the main text of the manuscript. We have added page numbers to the COREQ checklist to make this clearer as per the request from the editor. We have added further justifications or clarifications to certain items on the checklist as requested by reviewers (e.g. in response to reviewer comments 6, 8 and 9).

- 14. Overall I found the methods acceptable (with the small caveat that I had some small issues detailed above about missing details and small sample size). The study is well motivated – but the actual novelty and the overall contribution are limited in the current presentation. I feel that the authors could work on making more of the temporal analysis – and add something more critically reflective and possibly some sort of diagram to bring this aspect out more – as it is the three key themes on their own seem rather underwhelming and not new – but the longitudinal temporal analysis – of brought out more could be a valuable contribution.**

Author response:

Thank you. As detailed in response to reviewer comment 2 and 10, the manuscript now includes a diagram that illustrates the development of themes across the various time points. We also reflect upon the interrelated nature of the themes and the difficulties in separating individual and group level responses (see pages 16, 17 and 19).

Reviewer: 2

The manuscript explores the views and experiences of adolescents who participated in a gamified 24-week primarily school-based physical activity (PA) intervention (the StepSmart Challenge) in three schools in Belfast, Northern Ireland. The study is based on the analysis of focus groups. The main strength researchers report is that the study is based on a longitudinal qualitative study at four time-points (baseline, end of each of two intervention phases and 1 year follow-up). I think the study is relevant and will be interesting for a good number of researchers from different fields whose focus is on applying gamification among children and adolescents. Though I think the manuscript is good, I still have some comments:

- 15. 1) The term gamification is not defined. Though there is a large number of definitions one can find, the authors should include the definition they were using while designing BCT gamified elements for the StepSmart Challenge. What did they have in mind while designing it this way? What BCT gamified elements they found crucial for a successful experiment? They should clarify that. It would be helpful if the authors would state gamified elements they find important for applying gamification in health-related studies among adolescents (in the "Background" section), and add a column in Table 2 with these elements in the following section.**

Author response:

As noted in the introduction, some key gamification strategies, such as feedback on players' performance to allow them to set goals and monitor progression, competing with others, and use of incentives, are all evidence-based Behaviour Change Techniques (BCTs). Table 2 details the evidence-based BCTs incorporated in the intervention using a standardised taxonomy developed by Michie et al (2013). However, interventions that have used gamification elements have rarely been grounded in well-established theoretical frameworks and we know little about the views and experiences of participants. Briefly, our intervention was primarily informed by self-determination theory (Ryan and Deci, 2002) and self-regulation control theory framework (Carver and Scheier, 1982: 1998). Motivational messages (persuasion) and social support (vicarious experience) were also introduced to promote self-efficacy perceptions (Bandura, 1986). Gamification elements were used as a method to deliver the intervention rather than the intervention being explicitly underpinned by

gamification theory. We have changed the title to more accurately reflect that this was a gamified intervention rather than an intervention explicitly based on gamification theory.

16. 2) There is a discussion in gamification field about the role of intrinsic and extrinsic motivation. Gamification is about motivation anyway. But it is not clear what the authors expected in terms of intrinsic and extrinsic motivation and how they would be related in their study. They discuss intrinsic and extrinsic motivation in the “Discussion” part, but I was wondering how they tried to start with the extrinsically motivated PA behavior and change it to intrinsically motivated behavior, and if the gamified design of the StepSmart Challenge was successful with that.

Author response:

The intervention was phased so that extrinsic motivators such as prizes for ‘walker of the week’ were the primary driver of physical activity behaviour (Phase 1). However, this element of the intervention was embedded alongside intrinsically motivated evidence-based behaviour change techniques such as monitoring and feedback of behaviour. Phase 2 and 3 of the intervention purposefully reduced the emphasis from the reward element towards the importance of self-monitoring and feedback.

Further, we have developed Figure 1 which provides a thematic illustration of general (group-level) consensus over a 1 year period. The complexities contained within each pathway highlight the difficulty in developing a group-based PA intervention that will motivate all participants in a similar manner. It also illustrates the interrelated nature of the themes, and how experiences of one aspect of the intervention can influence other components. However, recognition-based incentives, the provision of feedback on performance, and opportunities for social connectivity were shown to be key gamification strategies with potential for motivating PA throughout the intervention. This is in line with SDT which posits that supporting the participants’ innate desires competence and their sense of relatedness with others could help achieve a higher quality of motivation that is long-lasting.

17. 3) The authors should state the research problem more clearly. Now there is only a descriptive analysis of focus groups at different time points, but it is not focused on the research problem. It would make sense if the authors would pose some research questions like “why gamification in PA intervention does not work as expected among some groups of adolescents” or “how gamification can be improved while applying to adolescents in health studies” or “how gamification can be helpful for those with low PA” or “how the long-term gamification effect can be maintained” or something else. A longitudinal qualitative study can help us better understand these issues which we can find in a quantitative study. Here we can explain or make suggestions based on the analysis of focus groups. A good number of experimental studies in different fields show that gamification can be efficient among both adults and children/adolescents. However the effect size varies a lot, and the stability of this effect varies in time (the effect size within the course of time), or we can not disentangle the effect of different gamified elements. I believe this qualitative study can help us answer some of the questions related to the duration of the gamification effect and the factors that have an effect on that duration, or effects of different gamified elements on different groups of adolescents, especially on those with a low level of PA. Again, there should be more problem-focused analysis of the materials the authors have. If researchers analyze temporal changes, then for instance they can report which gamified elements have the long-term effect in terms of motivation, which of them varied most, etc. With that being said, I would ask the authors to include some basic quantitative results in their study to what they report on p.6 (l.16-19), e.g. the effect size at different time points compared to the control condition + any other relevant results depending on the research problem they would focus on.

Author response:

Thank you. Please see our response to reviewer comment 4 where we have included research questions and study objectives.

The manuscript now includes a diagram (see Figure 1) that illustrates the development of themes across the various time points. A narrative summary is also provided and we have explained how this approach enabled the research team to gain additional insight and depth (see pages 16, 17 and 19).

18. 4) The authors analyze data thematically. They identified three core themes: competition; incentives; influence of friends. Though it makes sense, it is still confusing to have these particular themes without any explanation or theoretical foundation. I would suggest to have theoretically-driven themes according to the main research problem (see point 3 above). It might be the elements of SDT theory and the indicators of these elements in their study.

Author response:

Thank you for this suggestion. As noted in the reviewers comment we feel that each theme has been theoretically driven using SDT theory. We have made additional reference to this within manuscript in order to make clearer.

19. 5) In “Discussion” part the authors report some interesting findings but sometimes no explanation or hypotheses based on their analysis (e.g., long-term effect, lower interest – the balance between extrinsic and intrinsic motivation, etc.).

Author response:

A more in-depth explanation regarding the proposed underpinning theoretical pathways of behaviour change related to SDT is available within the main feasibility paper, which is currently under review with another journal. This manuscript can be provided to the reviewers on request if required. Briefly, the intervention was primarily informed by self-determination theory (Ryan and Deci, 2002) and self-regulation control theory framework (Carver and Scheier, 1982: 1998). Motivational messages (persuasion) and social support (vicarious experience) were also introduced to promote self-efficacy perceptions (Bandura, 1986).

20. 6) This part was not clear: “This could be linked to the provision of material incentives which was contingent on ‘winning’ the competition. Over the course of the intervention, material incentives continued to motivate some participants, suggesting that intrinsic motivation and autonomy were not undermined.” - How you can make such conclusions?

Author response:

We agree with the reviewer that this interpretation of the findings cannot be made from our study. We have re-written this sentence to remove the suggestion regarding undermining intrinsic motivation from the manuscript (page 17).

21. 7) When I was reading about the difference in individual and team competitions among girls and boys I was wondering if it is an overgeneralization based on the data they have (2 “all girls” school and 1 “all boys” school”). Again if this in line with the quantitative results they have, they should report that.

Author response:

We acknowledged this limitation on page 20, “Only three focus groups participated in the study, and all were single sex schools. Consequently the purported gender differences may be an artefact of differences in socio-economic status as well as or in addition to gender differences.” The issues regarding interpretation of the data and generalisability of the findings are the same for the quantitative and qualitative findings given the characteristics of the included schools and participants.

Some minor issues:

22. 8) The researchers have good numbers of those who participated in focus groups at each time point except T3 with having a problem with a particular school (school D). I think one should include that fact in the “limitations” section.

Author response:

This has now been added to the limitations section (page 20).

23. 9) Some components from the Table 2 are not analyzed or discussed for some reason (e.g, website) – these components did not have any effect in the study?

Author response:

The website was largely a delivery vehicle for a number of the BCTs. It is included in Table 2 to provide the reader with a detailed description of the intervention. However, it did not appear as a distinct theme or sub-theme in our thematic analysis and therefore has not been discussed in depth in the manuscript.

VERSION 2 – REVIEW

REVIEWER	Aigul Mavletova National Research University Higher School of Economics, Russia
REVIEW RETURNED	06-Mar-2018
GENERAL COMMENTS	The authors addressed all the comments and added important parts in the manuscript which is now clearer with the research goals and research outcomes.